# Identification of a new gibberellin receptor agonist, diphegaractin, by a cell-free chemical screening system

Akira Nozawa[1], Ryoko Miyazaki[1], Yoshinao Aoki[2], Reina Hirose[1], Ryosuke Hori[1], Chihiro Muramatsu[1], Yukinori Shigematsu[3], Keiichirou Nemoto[4], Yoshinori Hasegawa[5], Keiko Fujita[6], Takuya Miyakawa [7,8], Masaru Tanokura[7], Shunji Suzuki [2] & Tatsuya Sawasaki [1✉]

Gibberellin (GA) is a phytohormone that regulates various developmental processes during the plant life cycle. In this study, we identify a new GA agonist, diphegaractin, using a wheat cell-free based drug screening system with grape GA receptor. A GA-dependent interaction assay system using GA receptors and DELLA proteins from *Vitis vinifera* was constructed using AlphaScreen technology and cell-free produced proteins. From the chemical compound library, diphegaractin was found to enhance the interactions between GA receptors and DELLA proteins from grape in vitro. In grapes, we found that diphegaractin induces elongation of the bunch and increases the sugar concentration of grape berries. Furthermore, diphegaractin shows GA-like activity, including promotion of root elongation in lettuce and Arabidopsis, as well as reducing peel pigmentation and suppressing peel puffing in citrus fruit. To the best of our knowledge, this study is the first to successfully identify a GA receptor agonist showing GA-like activity in agricultural plants using an in vitro molecular-targeted drug screening system.

[1] Proteo-Science Center, Ehime University, 3 Bunkyo-cho, Matsuyama, Ehime 790-8577, Japan. [2] The Institute of Enology and Viticulture, University of Yamanashi, 1-13-1, Kitashin, Kofu, Yamanashi 400-0005, Japan. [3] Fruit Tree Research Center, Ehime Research Institute of Agriculture, Forestry and Fisheries, 1618 Shimo-idai, Matsuyama, Ehime 791-0112, Japan. [4] Iwate Biotechnology Research Center, 22-174-4 Narita, Kitakami, Iwate 024-0003, Japan. [5] Department of Applied Genomics, Kazusa DNA Research Institute, 2-6-7 Kazusa-kamatari, Kisarazu, Chiba 292-0818, Japan. [6] Faculty of Bioresource Sciences, Prefectural University of Hiroshima, 5562 Nanatsuka-cho, Shobara, Hiroshima 727-0023, Japan. [7] Department of Applied Biological Chemistry, Graduate School of Agricultural and Life Sciences, The University of Tokyo, 1-1-1 Yayoi, Bunkyo-ku, Tokyo 113-8657, Japan. [8] Present address: Graduate School of Biostudies, Kyoto University, Kitashirakawa-oiwakecho, Sakyo-ku, Kyoto 606-8502, Japan. ✉email: sawasaki@ehime-u.ac.jp

Gibberellin (GA) is a phytohormone produced by plants and some microorganisms, including fungi and bacteria, which regulates a wide variety of developmental processes throughout the plant life cycle[1]. Although GAs comprise a group of over 100 structurally related compounds containing an *ent*-gibberellane skeleton, only some GAs such as $GA_1$, $GA_3$, $GA_4$, and $GA_7$ (Fig. 1a) have biological activities[2]. $GA_1$, $GA_4$, and $GA_7$ are the major natural bioactive GAs, while $GA_3$ is a secondary metabolite from the fungus *Gibberella fujikuroi*[3]. The organic synthesis of natural bioactive GAs is made difficult by complex production processes[2]. $GA_3$ isolated from fungus has been widely utilised in the field of agriculture and horticulture as a plant growth regulator. $GA_3$ is therefore the only GA in which the manufacturing production method has been established via submerged fermentation.

As representative applications in agriculture, $GA_3$ is widely used to promote seed germination, stem elongation, flowering, fruit maturation, and the production of seedless fruits[4,5]. However, the effects of $GA_3$ often depend on the plant species[6]. In the case of seedless grapes, $GA_3$ is unable to induce the seedless trait in certain varieties. Although the reason for this has been elusive, a possible explanation could be catabolism by plant enzymes. In fact, 2,2-dimethyl $GA_4$, which is not catabolised by GA 2-oxidases, enzymes for GA inactivation, showed high and continuous activity of stem elongation in rice and maize[7]. Therefore, GA agonists with a structure different from GA may be effective even in these cases. As GA agonists, which do not have an *ent*-gibberellane skeleton, are still limited (AC-94377[8], 67D[9], helminthosporic acid analogue[10]), the development of new GA agonists with different structures from GA is expected in the global agricultural industry.

GA response in plants is initiated by the binding of bioactive GA to its receptor GID1 protein[11]. Upon GA binding, the conformation of GID1 changes, and the GID1-GA complex increases the ability to bind to DELLA proteins, which are negative regulators of GA signalling[12]. The DELLA proteins binding to the GID1-GA complex are degraded via the ubiquitin-proteasome pathway, and subsequently, the GA response is induced[12,13]. In a previous study, we reported that GA-dependent interactions between GID1 and DELLA proteins were detected by the AlphaScreen system[14]. This AlphaScreen-based interaction detection system could be easily diverted to high-throughput drug screening. In the present study, based on the information regarding the GA response elicited by the GID1-DELLA interaction, we attempted to identify GA receptor agonists using our GID1-DELLA interaction detection system as a screening system.

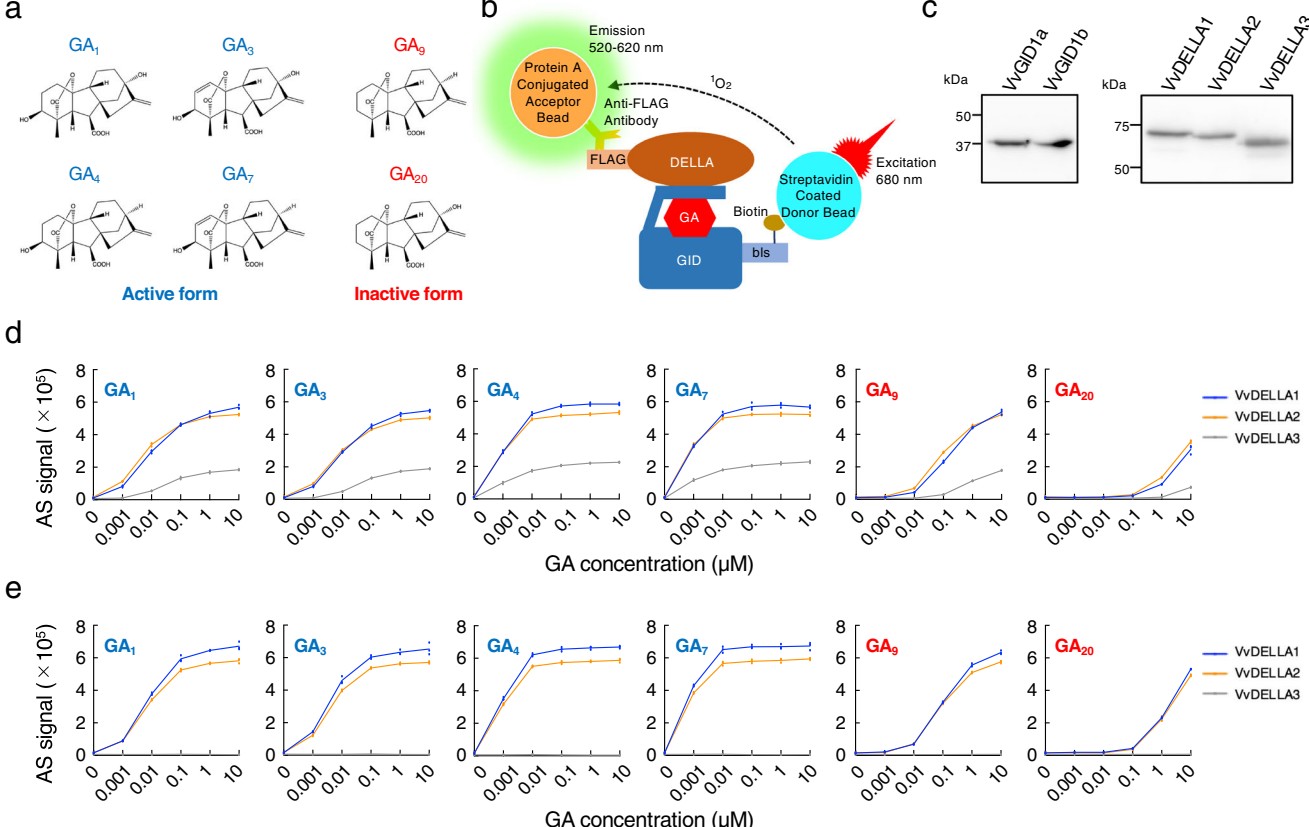

**Fig. 1 In vitro interaction analysis of GID1 and DELLA proteins from grapes. a** Structures of GAs. **b** Principle of interaction analysis between GID1 and DELLA using the AlphaScreen system. Biotinylated GID1 protein binds to streptavidin on donor beads with an extremely specific and high affinity. The protein A-coated acceptor beads were combined with FLAG-tagged DELLA protein using an anti-FLAG antibody. The GID1-DELLA complex forms a large complex with two types of beads through the antibody and streptavidin. After illumination at 680 nm, the donor bead converts ambient oxygen to singlet oxygen, and the singlet oxygen is transferred across to activate the acceptor bead and subsequently emits light at 520–620 nm. **c** Synthesis of VvGID and VvDELLA proteins. Biotinylated VvGID1 and FLAG-tagged VvDELLA proteins were synthesised using a wheat cell-free system. The synthesis of these proteins was confirmed by immunoblotting using anti-biotin and anti-FLAG antibodies. **d** Interaction assay between VvGID1a and VvDELLA proteins. The interaction of VvGID1a with VvDELLA proteins was analysed using the AlphaScreen system using GAs at various concentrations. Data are shown as mean and individual data points from three independent experiments. **e** Interaction assay between VvGID1b and VvDELLA proteins. The interaction of VvGID1b with VvDELLA proteins was analysed using the AlphaScreen system using GAs at various concentrations. Data are shown as mean and individual data points from three independent experiments.

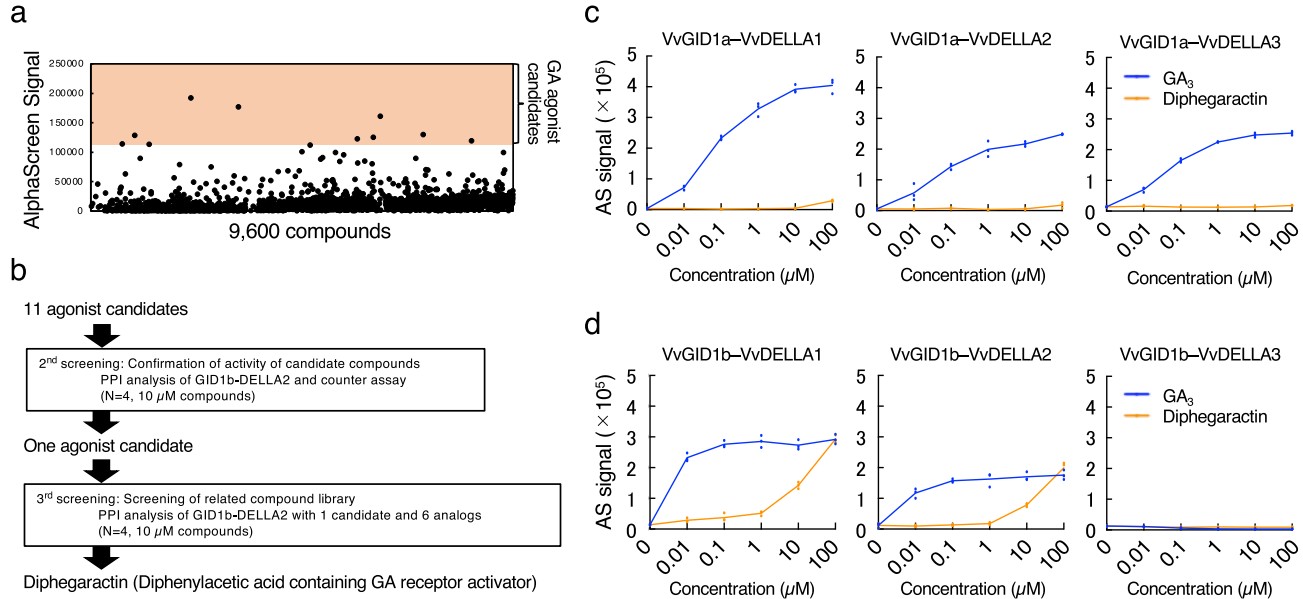

**Fig. 2 Isolation of a GA receptor agonist, diphegaractin, and its specific activity for VvGID1 and VvDELLA proteins. a** Result of GA receptor agonist screening using a chemical library containing 9600 chemical compounds. **b** Flow chart of GA receptor agonist isolation. **c** Dose-dependent activity of diphegaractin for interaction between VvGID1a and VvDELLA proteins. The interactions of VvGID1a with VvDELLA proteins were analysed by AlphaScreen system using $GA_3$ or diphegaractin at various concentrations. Data are shown as mean and individual data points from three independent experiments. **d** Dose-dependent activity of diphegaractin for interaction between VvGID1b and VvDELLA proteins. The interactions of VvGID1b with VvDELLA proteins were analysed by AlphaScreen system using $GA_3$ or diphegaractin at various concentrations. Data are shown as mean and individual data points from three independent experiments.

## Results

**Construction of a protein-protein interaction analysis system between GA receptors and DELLA proteins with GAs.** We previously identified several agonists for ABA receptors, and showed that agonist compounds prefer a receptor used in the screening[15]. From these results, we believe that the utilisation of a receptor protein from crops is a better approach for developing a useful agonist for agriculture, but not model plants, such as *Arabidopsis*. As such, we selected the grape *Vitis vinifera* L. cv. Pinot noir for the screening of GA agonists because the sequence of genes for GA receptors and DELLA proteins has been reported, as well as being a well-loved variety all over the world[16]. For screening, cDNA clones for two GA receptors (VvGID1a and VvGID1b) and three DELLA proteins (VvDELLA1, VvDELLA2, and VvDELLA3) were isolated from cDNAs made from the mRNA of grape leaves. The amino acid sequences of the proteins encoded by these clones and the phylogenetic trees of GA receptors and DELLA proteins from *Arabidopsis*, rice, grape, and lettuce are shown in Supplementary Figs. 1 and 2.

In a previous study, we demonstrated the GA-dependent interaction between GA receptors and DELLA proteins from *Arabidopsis* using the AlphaScreen system[14]. This system was applied to analyse the interaction of GA receptors with DELLA proteins from grapes (Fig. 1b). Biotinylated VvGID1 and FLAG-tagged VvDELLA proteins were synthesised using a wheat cell-free protein production system. As a result, it was confirmed that the levels of VvGID1 and VvDELLA proteins were almost identical (Fig. 1c). In total, six types of GAs, four active GAs ($GA_1$, $GA_3$, $GA_4$, and $GA_7$) and two inactive GAs ($GA_9$ and $GA_{20}$), were used for the interaction assay with these proteins. VvGID1a exhibited GA-dependent interactions with VvDELLA1-3 (Fig. 1d). Active GAs induced these interactions at concentrations greater than 1 nM (Fig. 1d). Although a much higher concentration was required, inactive GAs also induced these interactions (Fig. 1d). A similar AlphaScreen signal was detected in the interaction assay between VvGID1b and VvDELLA1 or VvDELLA2 (Fig. 1e). In contrast, the AlphaScreen signal in the assay of VvGID1a and VvDELLA3 was lower, indicating that the interaction affinity between VvGID1a and VvDELLA3 was lower than that of VvDELLA1 and VvDELLA2 (Fig. 1d). VvGID1b also exhibited active and inactive GA-dependent interactions with VvDELLA1 and VvDELLA2 in a similar manner (Fig. 1e). However, VvGID1b did not interact with VvDELLLLA3 in the presence of any tested GA (Fig. 1e). All VvGID1s interacted with VvDELLAs in a GA-dependent manner, with the exception of VvGID1b–VvDELLA3, similar to the results reported by Acheampong et al. for a yeast two-hybrid assay[16]. This indicates that the GA-dependent interaction of VvGID1 with VvDELLA is reproduced by our cell-free based interaction analysis system based on the AlphaScreen system.

**Screening of a GA receptor agonist using a wheat cell-free based chemical screening system.** By applying the cell-free based interaction analysis system, we attempted to identify GA receptor agonist compounds from the chemical library. As this system is able to directly use cell-free synthesised proteins for chemical screening without purification, it represents both a cost- and time-saving method for screening using large-scale chemical libraries. To identify these compounds, we screened a diverse set of 9600 synthesised chemical compounds established by the Drug Discovery Initiative (The University of Tokyo, Japan). Mono-biotinylated VvGID1b and FLAG-tagged VvDELLA2 were selected for screening because the expression of VvGID1b and VvDELLA2 was higher in developing fruit[16]. They were incubated in a 384-well plate containing individual chemicals at final concentrations of 10 $\mu$M, and the VvGID1b–VvDELLA2 interaction was analysed using the AlphaScreen system. If an agonist compound to the GA receptor was present, the interaction signal of VvGID1b–VvDELLA2 would increase. In the 1st screening using 9600 synthesised chemicals, we identified 11 agonist candidates (Fig. 2a). Next, reproducibility of the activity of these agonist candidates were confirmed as 2nd screening and at the same time we checked whether these agonist candidates increase

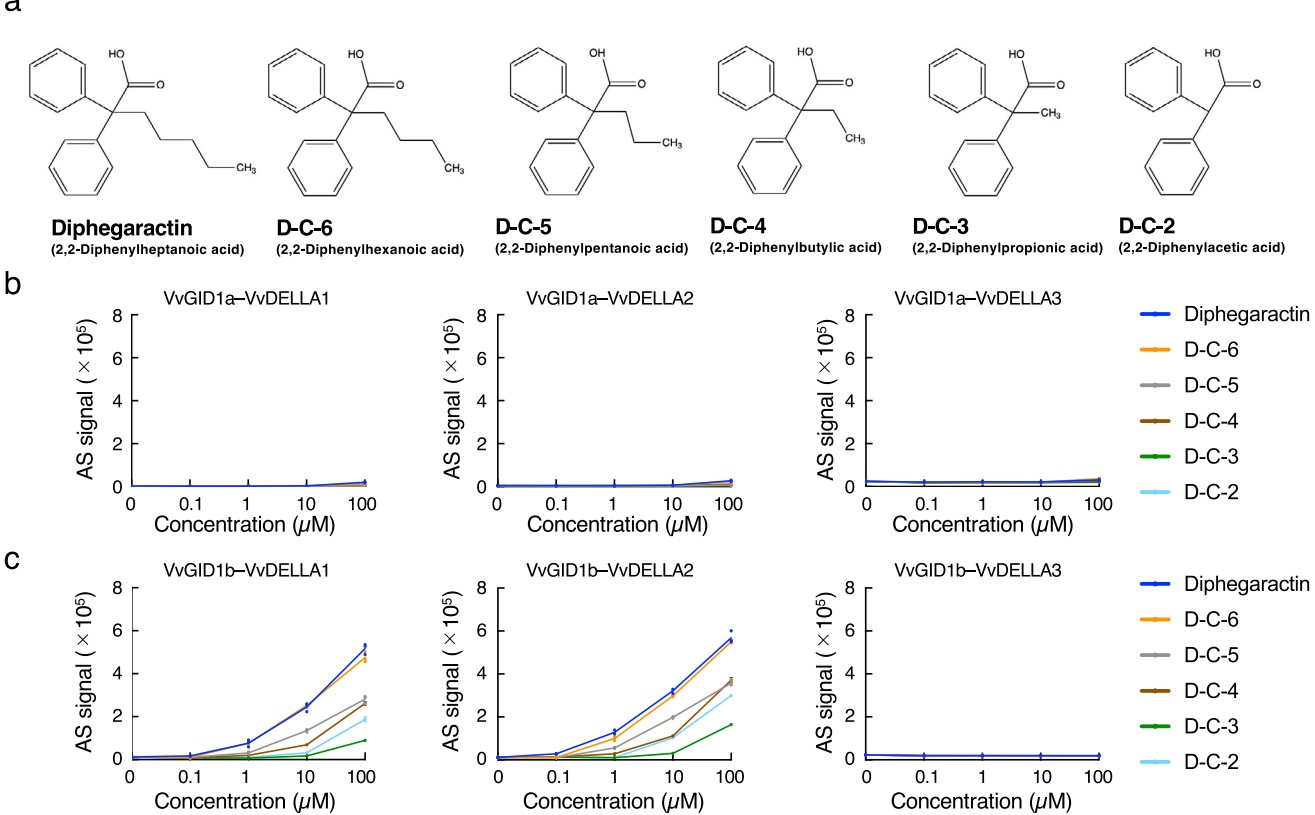

**Fig. 3 Comparison of GA receptor agonist activity among diphegaractin and analogue compounds. a** Structures of diphegaractin and its analogues.
**b** Dose-dependent activity of diphegaractin and its analogues for interaction between VvGID1a and VvDELLA proteins. Interaction of VvGID1a with VvDELLA proteins were analysed by AlphaScreen system using diphegaractin or its analogues at various concentrations. Data are shown as mean and individual data points from three independent experiments. **c** Dose-dependent activity of diphegaractin and its analogues for interaction between VvGID1b and VvDELLA proteins. Interaction of VvGID1b with VvDELLA proteins were analysed by AlphaScreen system using diphegaractin or its analogues at various concentrations. Data are shown as mean and individual data points from three independent experiments.

AlphaScreen signal independently of promoting interaction between VvGID1b and VvDELLA2 as counter assay (Fig. 2b). Further agonist candidate analogues were tested their agonist activity in 3rd screening (Fig. 2b). Finally, a single chemical compound, 2,2-diphenylheptanoic acid, which biochemically functions as a GA receptor agonist for VvGID1b, was named diphegaractin (diphenyl acetic acid containing GA receptor activator) (left compound in Fig. 3a). The structure of diphegaractin does not contain an *ent*-gibberellane skeleton, which is common to all gibberellins.

To evaluate the ability of diphegaractin to promote the interaction of VvGID with VvDELLA proteins, we analysed the interaction-promoting activity of diphegaractin at all combinations of VvGID and VvDELLA proteins. The results showed that diphegaractin induced two interactions, VvGID1b–VvDELLA1 and VvGID1b–VvDELLA2 (Fig. 2c, d). However, it did not facilitate the interaction of VvGID1a–VvDELLA1, VvGID1a–VvDELLA2, VvGID1a–VvDELLA3, and VvGID1b–VvDELLA3 (Fig. 2c, d). In particular, the same level of interaction signal was observed in the combination of VvGID1b–VvDELLA1 and VvGID1b–VvDELLA2 at 100 μM concentration of diphegaractin and $GA_3$ (Fig. 2d). These results indicated that diphegaractin has receptor selectivity and is expected to have GA agonist activity.

**In vitro analysis of GA receptor agonist activity using diphegaractin and its related chemicals.** Diphegaractin is a two-phenyl-ring-attached heptanoic acid (Fig. 3a). In order to assess the

effect of alkyl group length in diphegaractin on the interaction-promoting activity of VvGID and VvDELLA proteins, we investigated the interaction-promoting activity of VvGID and VvDELLA proteins by several diphegaractin analogous compounds containing different alkyl groups, from ethane to heptane (Fig. 3a). In all combinations between two VvGID1 and three VvDELLA proteins, these interaction-promoting activities declined in correlation with the shortness of the alkyl group (Fig. 3b, c). Therefore, diphegaractin showed the highest interaction-promoting activity. In the combination of VvGID1a or VvGID1b and VvDELLA3, no compounds showed interaction-promoting activity (Fig. 3b, c). These results indicate that the length of the alkyl group in diphegaractin is important for promoting the interaction between VvGID and VvDELLA proteins.

**GID1-DELLA interaction-promoting activity of diphegaractin in other plants.** Next, to determine whether diphegaractin shows GA agonist activity in other plant GA receptors, we analysed interaction-promoting activity using GID1 and DELLA proteins from rice, Arabidopsis, and lettuce. In our previous study, when a chemical, lenalidomide, which induces interaction of CRBN and its substarte PLZF, was used at concentrations where 20-fold or more signals than negative control was detected in AlphaScreen, we detected interaction of CRBN and PLZF in vivo[17]. Hence, we considered that the signals 20-fold or more than negative control is physiological meaningful in AlphaScreen. Using this criterion, although diphegaractin activated negligibly the interaction of

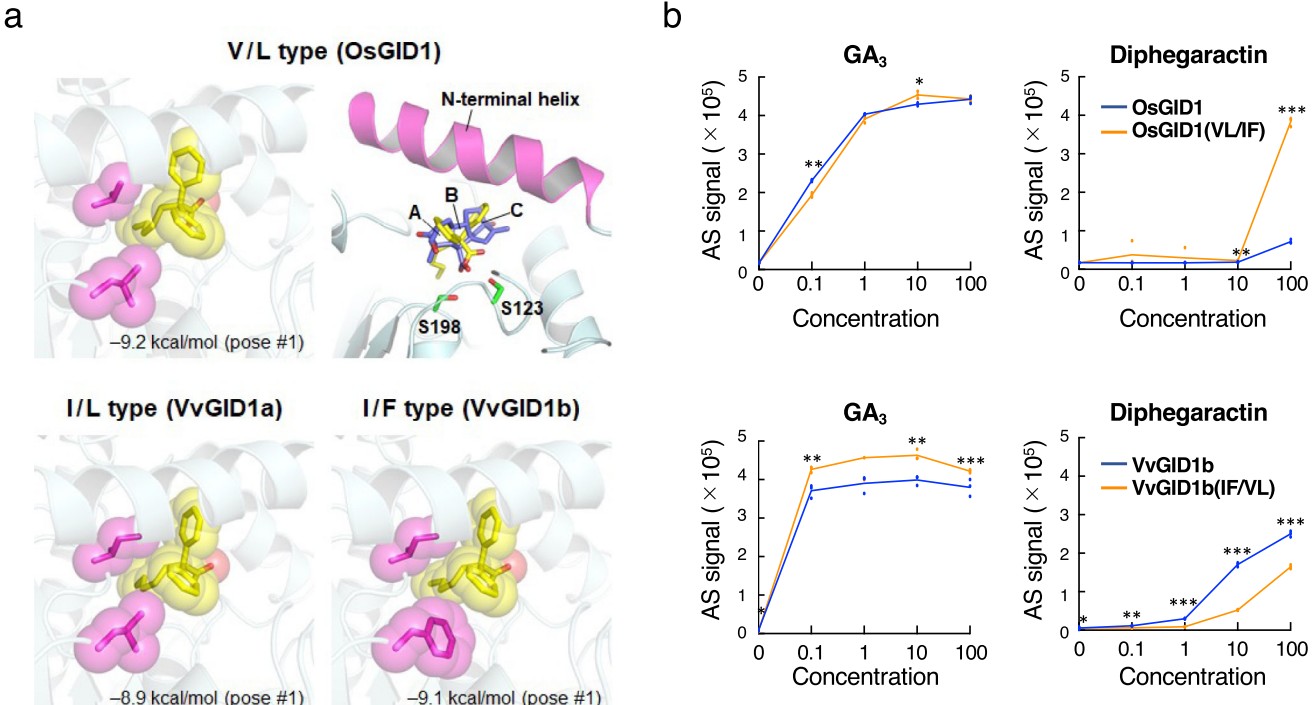

**Fig. 4 Putative action of diphegaractin toward three GID1 types. a** Docking models of diphegaractin in the GA-binding pockets of OsGID1, VvGID1a, and VvGID1b. Diphegaractin and GA₄ are represented by yellow and blue sticks, respectively. The three rings of GA₄ are labelled A, B, and C. Magenta sticks are the variable residues for the classification of GID1 types. Affinity score of the best docking pose (#1) is shown at the bottom of each docking model. **b** Effect of two amino acid residues, I319 and F323 of VvGID1b, for GA₃- or diphegaractin-dependent binding affinity between GID and DELLA proteins. Data are shown as mean and individual data points from three independent experiments. Asterisk indicates significant differences (*$P < 0.05$; **$P < 0.01$; ***$P < 0.001$, Student's $t$ test).

OsGID1-OsSLR (Supplementary Fig. 3), it promoted the interaction of AtGID1b with AtRGA, AtGID1c with AtRGA and AtGAI, LsGID1b-1 with LsDELLA1, and LsGID1b-2 with LsDELLA1 (Supplementary Figs. 4 and 5). These results indicate that diphegaractin is expected to have GA agonist activity at particular set of GID1 and DELLA proteins in some plants.

**Structural basis of diphegaractin recognition by GA receptors.** To understand the molecular basis of diphegaractin as a GA receptor agonist, we analysed the arrangements of diphegaractin in the GA receptor structure using docking models. The reported structures of GA₃/GA₄-bound OsGID1 show that V326 and L330 are located near GA₃ and GA₄ (Supplementary Fig. 6a)[18]. Both were conserved in AtGID1a, but the corresponding residue to V326 was substituted with Ile in other GID1s (Supplementary Fig. 6b). In addition, AtGID1b and VvGID1b had an additional substitution of Phe at the corresponding position to L330. All other residues in the GA-binding pocket were conserved among the GID1s. Therefore, GID1s were classified into three types based on the variation of the residues in the GA-binding pocket: V/L type for OsGID1 and AtGID1a, I/L type for AtGID1c and VvGID1a, and I/F type for AtGID1b and VvGID1b.

Next, we simulated docking models of diphegaractin by AutoDock Vina[19] using the crystal structure of OsGID1 (V/L type; PDB ID, 3ED1)[18] and the homology models of VvGID1a (I/L type) and VvGID1b (I/F type), which were automatically generated from the AtGID1a structure (PDB ID, 2ZSH)[20] by SWISS-MODEL server[21] with a high overall precision (0.91 GMQE and −1.45 QMEAN for VvGID1a; 0.89 GMQE and −1.51 QMEAN for VvGID1b). The number of docking poses was nine (V/L type), four (I/L type), and six (I/F type). The best docking

pose with the lowest affinity score (kcal/mol) adopted almost the same binding manner for the three types and occupied the binding site of GAs (Fig. 4a). The carboxy groups of diphegaractin and GA₄ were commonly oriented to conserved Ser residues (S123 and S198 for OsGID1) within the distance allowed to form hydrogen bonds. In addition, the diphenyl group mimicked the A and C rings of GA₄, suggesting that this binding mode is suitable for inducing a closed conformation of the N-terminal helix of GID1 for the interaction with DELLA, as well as GAs. We also simulated the docking models of diphegaractin analogous compounds containing alkyl groups of different lengths (Supplementary Fig. 7). The best docking poses were quite similar to those of diphegaractin, suggesting that the diphenyl and carboxy groups are the major components regulating the binding modes of diphegaractin and its analogous compounds. The comparative docking poses also support that the size of the alkyl group occupying the binding site contributes to the high binding affinity of diphegaractin (Supplementary Fig. 7). The variable residues of the three GID1 types were located near the diphenyl and alkyl groups of diphegaractin in the docking models (Fig. 4a). The bulkier residues of the I/F type appear to come into with diphegaractin more closely, and hence the residue types may affect the interaction with DELLA by modifying the binding affinity of diphegaractin toward GID1. Furthermore, we have conducted the docking simulation of the diphegalactin analogous compounds with a longer alkyl group than diphegalactin, 2,2-diphenyloctanoic acid (D-C-8), 2,2-diphenynonanoic acid (D-C-9), and 2,2-diphenyldecanoic acid (D-C-10). All the tested compounds showed the similar binding pose to diphegaractin toward the GA-binding pockets of OsGID1 (Supplementary Fig. 8). VvGID1a was also presumed to have a GA-binding pocket suitable to bind diphegaractin homologous compounds with a long acyl group since DC8 and DC10 showed

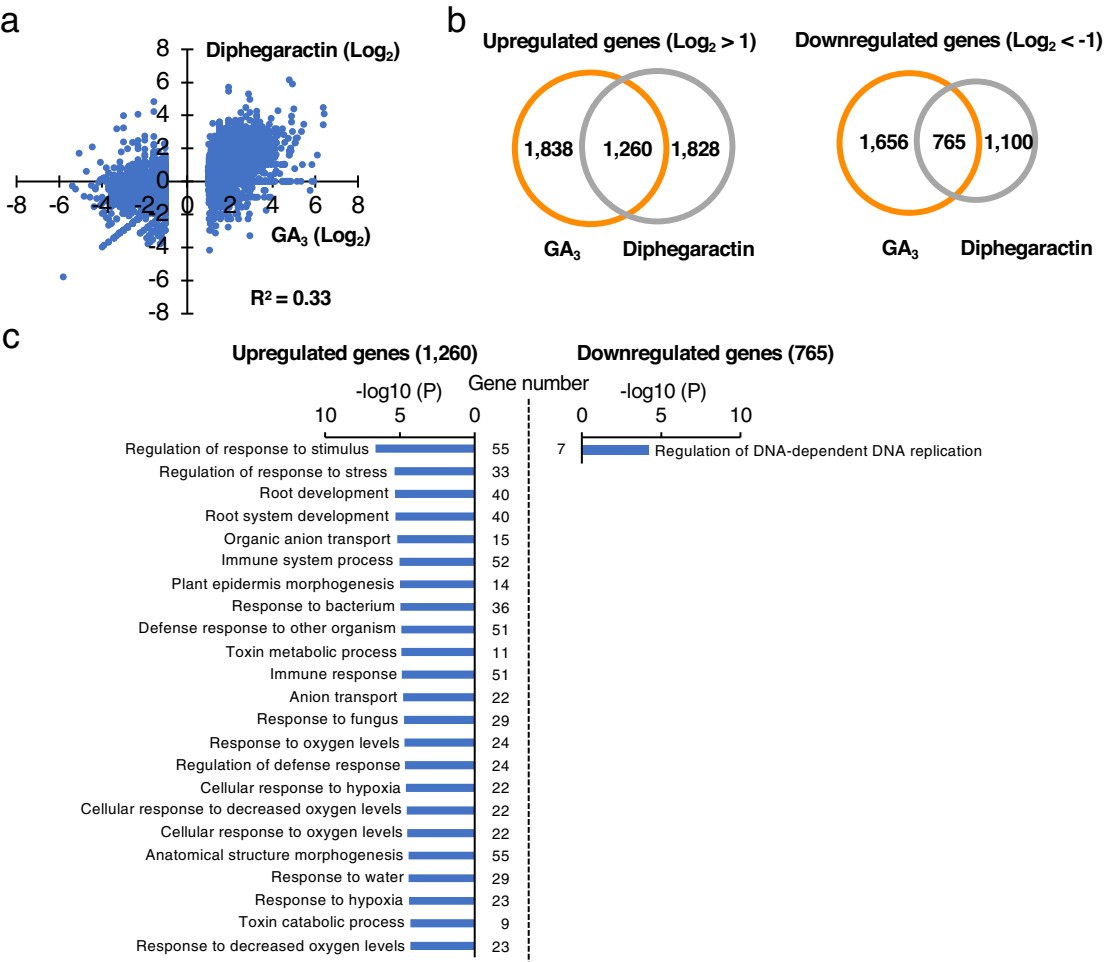

**Fig. 5 Effect of GA₃ and diphegaractin for gene expression in *Arabidopsis*. a** Scatter plot $\log_2$-transformed values of differential expression genes responsive to GA₃ (*y* axis) and diphegaractin (*x* axis) treatment relative to DMSO control. Dots represent upregulated ($\log_2$ fold change >1) and downregulated ($\log_2$ fold change < −1) genes by GA₃-treatment. The coefficient of determination ($R^2$) between GA₃ and diphegaractin was calculated and indicated in the figure. **b** Venn diagrams showing the overlap of upregulated ($\log_2$ fold change >1) and downregulated ($\log_2$ fold change < −1) genes between GA₃ and diphegaractin treatments. **c** Gene ontology analysis of differentially expressed genes. Gene ontological analysis of 1260 upregulated genes and 765 downregulated genes was performed. Enriched biological processes are listed based on their *p* values.

binding poses similar to diphegaractin for VvGID1a. In contrast, there is no binding pose of the tested compounds observed in the GA-binding pocket of VvGID1b. We analysed the size of the GA-binding pockets using CASTp 3.0 server[22], suggesting that the pocket size of VvGID1b (169 Å$^3$) was smaller than those of OsGID1 (208 Å$^3$) and VvGID1a (173 Å$^3$). Depending on the shape of the GA-binding pocket, homologous compounds with longer alkyl groups than difegaractin are thought to cause steric hindrance to the GA-binding pocket of VvGID1b. Finally, we analysed detail arrangement of diphegaractin in VvGID1b using docking model (Supplementary Fig. 9). Based on this binding model, I/F type residues form hydrophobic interactions with diphenyl group and a part of alkyl group of diphegaractin. The alkyl group is arranged to fill the space enclosed by I/F type residues and other residues located at the bottom of the GA-binding pocket. The binding model also suggests that S115 and R243 contribute to hydrogen bond formation with the carboxy group of diphegaractin.

To evaluate the two different residues of GA receptors for the recognition of diphegaractin, we analysed the interaction of DELLA proteins with GA receptors in which the residues were swapped. The VvGID1b I319V/F323L mutant showed decreased interaction with VvDELLA2 by diphegaractin, while the OsGID1 V326I/L330F mutant increased the interaction with OsSLR by

diphegaractin (Fig. 4b). In contrast, swapping had little effect on the interaction between DELLA proteins and GA receptors by GA₃ (Fig. 4b). These results indicate that the results of modelling were appropriate and that the two residues of I319 and F323 are important residues for the effective activity of diphegaractin in the formation of the VvGID1b–VvDELLA2 complex.

**Comparison analysis of gene expression profiling between GA and diphegaractin treatments**. As the interaction between GID1 and DELLA proteins from *Arabidopsis* were promoted by diphegaractin, biological activity of diphegaractin was tested in *Arabiodpsis*. As shown in Supplementary Fig. 10a, the promotion of root growth by diphegaractin was observed in *Arabidopsis*. Since the biological activity of diphegaractin was confirmed, RNA sequence was next performed to verify the effect of diphegaractin for gene expression. *Arabidopsis* seedlings treated with GA₃ or diphegaractin were harvested, and transcriptome of these samples was analysed by RNA sequencing. A slightly correlated response was observed among the genes induced or repressed by GA₃ ($R^2 = 0.33$, Fig. 5a). Compared with the DMSO-treatment, 3098 and 3088 genes were induced more than twofold by GA₃- or diphegaractin-treatment, respectively, of which 1260 genes were

common in samples treated with $GA_3$ and diphegaractin (Fig. 5b). Conversely, 2421 and 1865 genes were suppressed by over twofold in $GA_3$- or diphegaractin-treatment, respectively, and 765 genes were suppressed in both $GA_3$- and diphegaractin-treatment (Fig. 5b). The gene ontology analysis of genes upregulated by both $GA_3$ and diphegaractin indicated that the regulation of response to stimulus, regulation of response to stress, root development, and root system development are higher-enriched biological processes (Fig. 5c). On the other hand, regulation of DNA-dependent DNA replication is only an enriched biological process in the analysis of downregulated genes (Fig. 5c). These results indicate that there are many genes whose expression is affected by both diphegaractin and $GA_3$ and that diphegaractin may have similar biological functions as $GA_3$ in various situations, including root development.

We next investigated effect of diphegaractin for expression of GA synthetic and GA metabolic genes. In this RNA sequencing experiment, GA synthetic genes downregulated ($\log_2$ fold change < −1) by $GA_3$-treatment were $AtGA20ox2$ (−3.9), $AtGA20ox3$ (−2.3), $AtGA3ox1$ (−2.0), and $AtGA3ox2$ (−1.6) and GA metabolic genes upregulated ($\log_2$ fold change >1) by $GA_3$-treatment were $AtGA2ox2$ (1.1) and $AtGA2ox6$ (2.9). In these genes, $AtGA20ox3$ was also similar level downregulated by diphegaractin (−2.3), indicating feedback regulation of $AtGA20ox3$ by GA would be mediated by AtGID1b/AtRGA, AtGID1c/AtRGA, and/or AtGID1c/AtGAI. Whereas, expression level of $AtGA20ox2$, $AtGA3ox1$, $AtGA3ox2$, and $AtGA2ox6$ were hardly affected by diphegaractin-treatment ($1 > \log_2$ fold change > −1). In contrast, $AtGA2ox2$ was upregulated by GA3 (1.1) but downregulated by diphagaractin (−3.0). These results indicate that the expression levels of most of GA synthetic and metabolic genes seem not to be affected by diphegaractin. The response of these genes to GA would be mediated by the combinations of GID1 and DELLA proteins except for AtGID1b/AtRGA, AtGID1c/AtRGA, and AtGID1c/AtGAI.

**Analysis of biological functions of diphegaractin in agricultural plants**. Gene expression analysis revealed many genes exist whose expression is controlled similarly by both $GA_3$ and diphegaractin (Fig. 5), suggesting that a part of GA agonist activity is attributed to diphegaractin. Since it did not affect the rice GA receptor (Supplementary Fig. 3b), we focused on eudicots. Thus, we investigated the biological function of diphegaractin in three families of agricultural plants: grape (Vitaceae), lettuce (Asteraceae), and orange (Rutaceae).

First, we investigated the effect of diphegaractin in grapes, because it was selected by chemical screening using grape GA receptor. The most popular biological activity of GA is its growth-promoting activity. Therefore, we investigated the effect of diphegaractin on the bunch length of three different grape cultivars (Kyoho, Shine Muscat, and Muscut Bailey A). Grape spikes from several cultivars were treated with diphegaractin or $GA_3$ before and after flowering. As shown in Fig. 6a, b, the growth-promoting activity of bunch length was observed in cultivars Kyoho, Shine Muscat, and Muscut Bailey A. In addition to growth-promoting activity, treatment with diphegaractin increased the sugar concentration in grape berries (cultivar Shine Muscat) (Fig. 6c). Parthenocarpy was observed in $GA_3$-treated grape berries but not in diphegaractin-treated grape berries (Supplementary Fig. 11). This result indicates that parthenocarpy may not be mediated by VvGID1b, although the expression of VvGID1b is high in developing fruits.

The growth-promoting activity of diphegaractin was further investigated in lettuce. As shown in Fig. 6d, although growth-promoting activity of diphegaractin for root length was less than

$GA_3$, the growth of lettuce roots was also promoted by diphegaractin.

In satsuma mandarin (Citrus unshiu), a Japanese orange, $GA_3$ is known to delay peel pigmentation and reduce peel-puffing activity. To test these activities, immature satsuma mandarin fruits were treated with diphegaractin or $GA_3$. After 34 days, the peel colour of the control fruits turned orange, while that of $GA_3$-treated fruits remained green. The peel colour of diphegaractin-treated fruits was between that of the control and $GA_3$-treated fruits (Fig. 6e). In addition, weak peel-puffing-reducing activity was also observed in diphegaractin-treated fruits 71 days after treatment (Fig. 6f).

Taken together, these results indicate that diphegaractin has solid GA-like biological activities in several agricultural plants, although the activity of diphegaractin may be narrower and weaker than that of $GA_3$. To our knowledge, this study is the first to successfully identify a GA receptor agonist using an in vitro molecular-targeted drug screening system and working in agricultural plants.

## Discussion

In this study, we used a protein-protein interaction analysis system based on the wheat cell-free system and AlphaScreen technology as a screening system for GA agonists. In this screening, two cell-free reaction mixtures, including translational GID1 and DELLA protein, were used without purification. These proteins and AlphaScreen beads with antibodies were applied to 384-well plates containing a compound library using an autodispenser. This screening system, which is less laborious and time consuming, allows for the chemical screening of GA agonists, including the preparation of proteins and compound screening, and was conducted for only two days. In particular, the screening of 9600 compounds using AlphaScreen technology was performed for only three hours. Although compounds isolated by this in vitro screening system need to be verified by subsequent in vivo analysis, these results suggest that this screening method is highly useful for the identification of candidate chemicals from compound libraries.

In contrast to GA, diphegaractin facilitates the interaction of particular pairs of GA receptors and DELLA proteins (Fig. 2c, d, Supplementary Figs. 3b, 4b, 5b). This selectivity of diphegaractin would be useful for investigating the biological roles of a GA receptor. In grapes, diphegaractin had a specific effect on VvGID1b (Fig. 2c, d). The promotion of bunch length and sugar concentration of berries was observed in diphegaractin-treated grapes (Fig. 6a–c). These processes are considered to be mediated by VvGID1b. In contrast, parthenocarpy was not observed in diphegaractin-treated grape berries (Supplementary Fig. 11), and VvGID1b may not be related to parthenocarpy. In Arabidopsis, AtGID1b was reported to express higher level in roots[23]. In addition, root growth of AtGID1b mutant was inhibited by lower concentration of Ancymidol, an inhibitor of GA synthesis, than that of AtGID1a and AtGID1c[24]. Furthermore, lettuce was reported to have high GA sensitivity in roots[25], and LsGID1b-1 and LsGID1b-2 were also reported to express preferentially in roots[24,26]. These results indicated AtGID1b, LsGID1b-1, and LsGID1b-2 contribute GA-induced root elongation. In this study, diphegaractin promoted interaction of AtGID1b, LsGID1b-1, and LsGID1b-2 with DELLA protein and root growth in Arabidopsis and lettuce (Supplementary Fig. 4b, 10a). These results suggest that AtGID1b, LsGID1b-1, and LsGID1b-2 would be mainly contributed to GA-induced root growth in Arabidopsis and lettuce, respectively. These results suggest that a receptor-specific agonist such as diphegaractin would be a powerful tool for understanding the biological roles of the receptor, especially when the receptor constitutes a family.

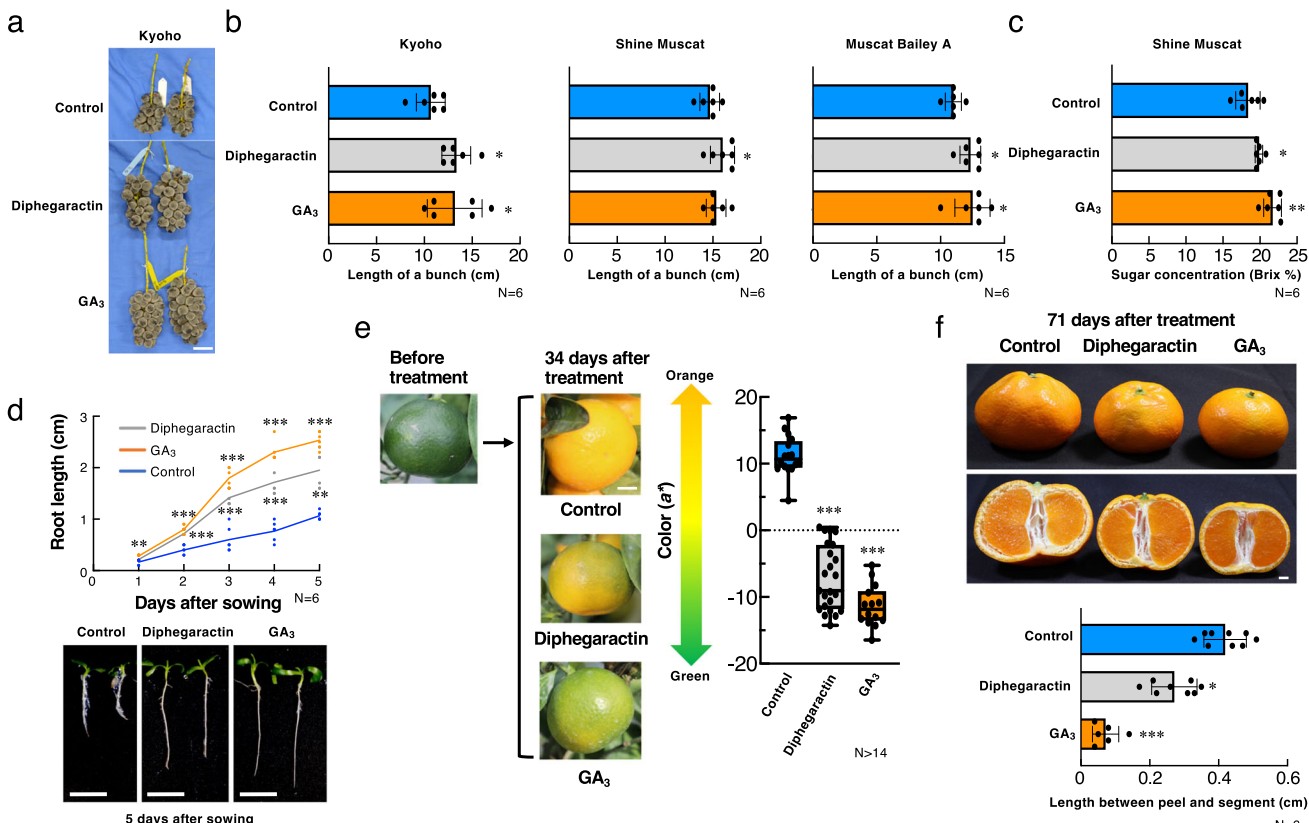

**Fig. 6 Biological analysis of diphegaractin in grapes, lettuce, and citrus. a, b** Effect of diphegaractin on length of bunches of grapes. Grape spikes were treated with GA₃ or diphegaractin. After 74 days, the bunch length was measured. Data are shown as mean ± standard deviation and individual data points from six independent experiments. Scale bar = 5 cm. **c** Effect of diphegaractin on sugar concentration of grape berries. Grape spikes were treated with GA₃ or diphegaractin. After 74 days, the sugar concentration of grape berries was measured. Data are shown as mean ± standard deviation and individual data points from six independent experiments. **d** Effect of diphegaractin on root length of lettuce. Lettuce seeds were sown on 1/2 MS agar plates supplemented with GA₃ or diphegaractin. The primary root length was measured at 1, 2, 3, 4, and 5 days after sowing. Data are shown as mean and individual data points from six independent experiments. Scale bar = 1 cm. **e** Effect of diphegaractin on colour of citrus peel. The fruits of satsuma mandarin treated with GA₃ or diphegaractin. After 34 days, the colour of the fruit peels was measured. Data are shown as mean ± standard deviation and individual data points from 14 or more independent experiments. Scale bar = 1 cm. **f** Effect of diphegaractin on peel-puffing. The fruits of satsuma mandarin treated with GA₃ or diphegaractin. After 71 days, the distance between the peel and segment was measured. Data are shown as mean ± standard deviation and individual data points from six or more independent experiments. Scale bar = 1 cm. Asterisk indicates significant differences from control (*P < 0.05; **P < 0.01; ***P < 0.001, Student's t test).

We showed that the amino acid residues of I319 and F323 in the GA-binding pocket of VvGID1b are important residues for the binding of diphegaractin with VvGID1b from docking model simulations and amino acid-residue swapping experiments (Fig. 4a, b). We classified GID1s into three types based on the variation in the residues in the GA-binding pocket (Fig. 4a). In this study, we tested the interaction-promoting activity of diphegaractin using nine kinds of GID1 proteins. The results showed that the interaction of four I/F type proteins (VvGID1b, AtGID1b, LsGID1b-1, and LsGID1b-2) with DELLA proteins was promoted by diphegaractin. However, the interaction-promoting activity of diphegaractin was not observed in the combination of two V/L type proteins (OsGID1 and AtGID1a) with DELLA proteins (Fig. 2d, Supplementary Fig. 3b, 4b, 5b). Furthermore, in the case of I/L type proteins, AtGID1c was induced to interact with DELLA proteins by diphegaractin. However, this effect was not observed in VvGID1a and LsGID1a (Fig. 2c, Supplementary Fig. 4b, 5b). These results suggest that GID1 proteins of the interaction between DELLA proteins are promoted by diphegaractin. Voegele et al.[27] reported that GID1 proteins in angiosperm fall into three phylogenetic groups: GID1ac eudicot group, GID1b eudicot group, and GID1 monocot group. We analysed the amino acid sequences of 36 GID1 proteins they used for

phylogenetic analysis; 37 GID1 proteins were used for the analysis, but we eliminated Allium cepa GID1 (BN001199) because of partial sequence[27]. The results showed that all 13 GID proteins belonging to the GID1b eudicot group were I/F type proteins and I/F type proteins were not found in the GID1ac eudicot and GID1 monocot groups. In addition, I/L type proteins were found in both the GID1ac eudicot group (10 out of 14) and GID1 monocot groups (3 out of 8). Taken together, diphegaractin may be effective for the GID1b eudicot group and a part of the GID1ac eudicot group and GID1 monocot group.

GA agonist activities of diphegaractin were observed in several plant species in this study (Fig. 6 and Supplementary Fig. 10). Among these, although results from lettuce and Arabidopsis were obtained in the laboratory (Fig. 6d, Supplementary Fig. 10a, b), those from grapes and satsuma mandarin were obtained in the field (Fig. 6a–c, e, f,). These results indicate that diphegaractin has the potential to be used as a GA agonist under natural conditions. At the moment, although the GA agonist activities of diphegaractin may be weaker than those of GA₃, as diphegaractin showed the same level of GID1s-DELLAs interaction-promoting activity with GA₃ at concentrations higher than 100 μM (Fig. 2d), stronger GA agonist activity may be observed at concentrations higher than those used in this study. In addition, the structure of

diphegaractin does not have an *ent*-gibberellane skeleton and is quite different from that of GAs. Hence, diphegaractin may have an advantage in that diphegaractin is not metabolised by the GA-inactivating pathway in plant bodies. Although further optimisation of the application procedure may be required, these findings suggest that diphegaractin would exert higher GA agonist activity in the field.

At present, several GA agonists without an *ent*-gibberellane skeleton have been reported, including phthalimide compounds, anthracene-derivative compounds, and helminthosporic acid analogues[8–10]. Diphegaractin is a diphenyl acetic acid-derivative compound and has a structure that is quite different from that of GA agonists. Furthermore, given the different specificities of GA agonist activity for receptors[8,9], the structure of diphegaractin would be a new lead compound for the development of subtype-specific GA agonists. In addition, diphegaractin has a relatively simple structure and can be synthesised chemically, suggesting that diphegaractin has characteristics suitable for lead compounds. We expect that next-generation GA agonists will be developed as plant growth regulators for agricultural crops.

## Methods

**Plant material**. *Vitis vinifera × V. labrusca* cvs. Kyoho, Shine Muscat, Muscat Bailey A, *Lactuca sativa*, *Citrus unshiu*, and *Arabidopsis thaliana* ecotype Columbia (Col-0) were used in this study. Kyoho, Shine Muscat, Muscat Bailey A, and *Citrus unshiu* were grown in the field. *L. sativa* and *A. thaliana* were grown in a growth cabinet.

**Chemicals**. Chemical compounds for screening were provided by the Drug Discovery Initiative at the University of Tokyo (Tokyo, Japan). $GA_1$ (OlChemIm, Olomouc, Czech Republic), $GA_3$, $GA_4$ (OlChemIm), $GA_7$ (OlChemIm), $GA_9$ (OlChemIm), $GA_{20}$ (OlChemIm), diphegaractin (Enamine, Monmouth Jct, NJ), D-C-6 (Enamine), D-C-5 (Toronto Research Chemicals, North York, Canada), D-C-4 (Sigma Aldrich Japan, Tokyo, Japan), D-C-3 (Sigma Aldrich Japan), and D-C-2 (Sigma Aldrich Japan) were prepared as stock solutions of 100 mM in DMSO, and appropriately diluted immediately before use. The final concentration of DMSO in the culture medium or assay buffer was 1% or less.

**Cell-free protein synthesis**. All proteins were synthesised by the wheat germ cell-free protein synthesis system using the WEPRO1240 expression kit (CellFree Sciences, Matsuyama, Japan) according to the manufacturer's instructions. Biotinylation at the biotin ligation site was carried out enzymatically using BirA biotin ligase[28]. The aliquots were used for expression analysis and interaction analysis.

**Immunoblot analysis**. Proteins synthesised by cell-free system were subjected to sodium dodecyl sulphate-polyacrylamide gel electrophoresis and transferred to a polyvinylidene difluoride membrane (Merck Millipore, Billerica, MA). The membrane was blocked using 5% skim milk in TBST (150 mM NaCl, 20 mM Tris-HCl (pH 7.5), 0.05% Tween-20) for 1 h. After blocking, the membrane was treated with horse radish peroxidase-conjugated anti-biotin antibody (Cell Signaling Technology, Danvers, MA, #7075, 1:5000) or anti-FLAG antibody (Sigma Aldrich Japan, A8592, 1:5000). The synthesised proteins were detected by using Immobilon Western Chemiluminescent HRP Substrate (Merck Millipore) according to the manufacturer's instruction. The signals were visualised with an Image Quant LAS 4000 mini (GE Healthcare, Buckinghamshire, UK).

**Interaction analysis of GA receptors with DELLA proteins**. For the protein-protein interaction analysis of GA receptors and DELLA proteins in the presence of GA or GA agonists, we prepared C-terminal biotinylated GA receptors and C-terminal FLAG-tagged DELLA proteins using a cell-free protein synthesis system. Protein-protein interaction analysis was performed using the AlphaScreen system. Briefly, 15 µl of a reaction mixture containing AlphaScreen buffer (100 mM Tris-HCl (pH 8.0), 0.1% Tween-20, and 1 mg/ml bovine serum albumin), 0.5 µl of biotinylated GA receptors, 0.5 µl of C-terminal FLAG-tagged DELLA proteins, and various concentrations of GAs or GA agonists were added to a 384-well Optiplate (PerkinElmer Japan, Yokohama, Japan). After incubation at 26 °C for 1 h, 10 µl of a detection mixture containing AlphaScreen buffer, 0.1 µl of streptavidin-coated donor beads (PerkinElmer Japan), 0.1 µl of protein A-coated acceptor beads (PerkinElmer Japan), and 5 µg/ml anti-FLAG M2 antibody (Sigma Aldrich) was added to each well. Thereafter, the plate was incubated for an additional hour. Luminescence signals were detected using an Envision plate reader (PerkinElmer Japan). The experiment was repeated three times. Data are presented as average values.

**Chemical library screening**. For chemical library screening, we used a core library containing 9600 synthesised chemical compounds established by the Drug Discovery Initiative (The University of Tokyo, Tokyo, Japan). All chemical compounds dissolved in DMSO were placed on a 384-well Optiplate (250 nl/well) at a concentration of 1 mM. Then, 15 µl of the reaction mixture containing GA receptors and DELLA proteins was dispensed into each well using a FlexDrop dispenser (PerkinElmer Japan). Each 384-well plate contained 32 negative control wells (containing biotinylated GA receptors and FLAG-tagged DELLA proteins) and 32 positive control wells (containing biotinylated GA receptors, FLAG-tagged DELLA proteins, and 10 nM $GA_3$). After incubation at 26 °C for 1 h, 10 µl of the detection mixture was added to each well using a FlexDrop dispenser, followed by incubation at 26 °C for 1 h (final concentrations: 10 µM each chemical compound, 1% DMSO). Luminescence signals were detected using an EnVision plate reader.

**Docking simulation**. The homology models of VvGID1a and VvGID1b were generated using the SWISS-MODEL server[20] using residues 6–342 as a query sequence. Chain A of the AtGID1a-$GA_3$ structure (PDB ID, 2ZSH) was automatically selected as a template model by the server. Two models were generated for VvGID1a, and the model with the highest GMQE score was used for the docking simulation. Polar hydrogens and charges were added to the crystal structure of OsGID1 (PDB ID, 3ED1, chain A) and the homology models using AutoDockTools-1.5.6 (The Scripts Research Institute). 3D models of diphegaractin and its analogues were obtained from PubChem (National Center for Biotechnology Information). 2,2-Diphenylbutylic acid (D-C-4) was created using the PRODRG sever[29]. UCSF Chimera 1.14[30] was used to add hydrogen and charge to all the ligand compounds and to minimise the models energetically. We selected the GA-binding pocket as a docking site, and the grid box was set to wrap around the pocket for each protein model. Docking simulation was performed using AutoDock Vina[31] with the following parameters: exhaustiveness, 8; number of modes, 100; and energy range, 3. First, we confirmed that the docking simulation worked well using $GA_4$ as a ligand. Next, diphegaractin or each analogous compound was applied to generate the docking models, which were automatically ranked based on the calculated affinity score (kcal/mol) by AutoDock Vina. The PyMOL Molecular Graphics System (version 2.4.0 Schrödinger, LLC) was used to depict all the structures.

**RNA sequence analysis and gene ontology biological process enrichment analysis**. The *Arabidopsis* seeds were sown on 1/2MS plate containing 10 µM paclobutrazol with or without 50 µM $GA_3$ or diphegaractin. After 10 days, the *Arabidopsis* seedlings were harvested. Total RNA was extracted using the RNeasy Plant Mini Kit (Qiagen, Hilden, Germany). Sequencing libraries were prepared according to the manufacturer's instructions for the Quant Seq 3′ mRNA-seq library preparation kit (Illumina, Lexogen, Vienna, Austria). Briefly, poly A RNA was purified from 200 ng of total RNA per sample using oligo-dT magnetic beads. The libraries were PCR-amplified for 13 cycles and purified with AMPure XP beads. Sequencing of the libraries was conducted on the Illumina NextSeq 500 system with single-end 75 bp reads. The raw reads were subjected to adapter trimming and quality trimming, followed by mapping to the *A. thaliana* genome (TAIR10) using the CLC Genomics Workbench (v20 QIAGEN) with default settings. After DeSeq normalisation, we analysed gene expression profiles across the samples. Differentially expressed genes responsive to both Diphegaractin and $GA_3$ treatments (1,260 upregulated and 765 downregulated genes, $\log_2$ fold change >1 or <−1) were annotated by the *Arabidopsis* Genome Initiative (AGI) locus codes, and were analysed for gene ontology. Gene ontology biological process enrichment analysis was performed using the PANTHER Classification System analysis tool on the homepage of the GOC website (http://geneontology.org). After gene ontology analysis, biological processes consisting of 50–999 annotated genes were selected. Biological processes with scores more than 4.2 (−log10 (*p* value)) are shown in Fig. 6c.

**Bunch length assay of the grapes**. For the bunch length assay, grape spikes were immersed in a treatment solution (0.1% DMSO, 0.1% Silwet L-77) supplemented with or without 73 µM (25 mg/l) $GA_3$, or 82 µM (25 mg/l) diphegaractin before and after flowering at 1-week intervals. After treatment, the grapevines were grown in the field. The bunch length was measured at 74 days after the 2nd treatment.

**Sugar concentration assay of grapes**. For the sugar content assay, grape spikes were immersed in a treatment solution (0.1% DMSO, 0.1% Silwet L-77) supplemented with or without 73 µM (25 mg/l $GA_3$, or 82 µM (25 mg/l) diphegaractin before and after flowering at 1-week intervals. After treatment, the grapevines were grown in the field. The sugar content was measured at 74 days after the 2nd treatment.

**Root length assay of lettuce**. For the root length assay, lettuce seeds were surface sterilised with 70% ethanol and washed three times with sterile water. The seeds were sown on half-strength Murashige and Skoog agar plates (2% sucrose) supplemented with or without 100 nM $GA_3$ or 50 µM diphegaractin. The plates were incubated vertically at 23 °C with a photoperiod of 16 h. The primary root length was measured at 0, 1, 2, 3, 4, and 5 days after sowing.

**Peel colour assay of satsuma mandarin.** For the peel colour assay, small fruits of satsuma mandarin were immersed in a treatment solution (0.5% DMSO, 0.45% Silwet L-77) supplemented with or without 100 nM GA$_3$ or 5 μM diphegaractin. After treatment, satsuma mandarin trees were grown in a plastic greenhouse. The colour (lightness ($L^*$), redness ($a^*$), and yellowness ($b^*$)) of fruit peels was measured using a chroma metre (Spectrophotometer CR-300; Minolta, Osaka, Japan) at 34 days after treatment. The $a^*$ value was used as the pigmentation index from green to orange.

**Peel-puffing assay of satsuma mandarin.** For the peel-puffing assay, small fruits of satsuma mandarin were immersed in a treatment solution (0.5% DMSO, 0.45% Silwet L-77) supplemented with or without 100 nM GA$_3$ or 5 μM diphegaractin. After treatment, satsuma mandarin trees were grown in a plastic greenhouse. The distance between the peel and the segment was measured with a calliper at 71 days after treatment.

**Statistics and reproducibility.** The results are expressed as means as mean ± standard deviation for the indicated number of observations. We used Student's $t$ test for pairwise analysis and Tukey's test for comparing multiple samples. The significant levels were indicated as star numbers: $^*p < 0.05$, $^{**}p < 0.01$, $^{***}p < 0.001$.

**Reporting summary.** Further information on research design is available in the Nature Portfolio Reporting Summary linked to this article.

## Data availability
Source data behind the graphs are available as Supplementary Data 1. Uncropped and unedited blot images are available as Supplementary Fig. 12. RNA sequencing data were deposited to DNA Data Bank of Japan (Accession number: DRA015789, DRA015790, DRA015791). All other data are available from the corresponding author (or other sources, as applicable) on reasonable request.

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

## Acknowledgements
We would like to thank Tetsunan Yamamoto and Yuta Kobayashi, University of Yamanashi, for their professional maintenance of grapevines, all staff of the Fruit Tree Research Center (Ehime Research Institute of Agriculture, Forestry and Fisheries) for plant management, the Drug Discovery Initiative at the University of Tokyo for providing the chemical libraries, Di Mu, Riku Nariyama, Shuntaro Oishi, Michihito Ono, Jumpei Sakai, Prefectural University of Hiroshima, Saya Matsuoka, Souta Shinohara, Yuki Shoya, So Tokunaga, Ehime University, for supporting the measurement of grapevine samples, Katsunari Maruyama, Ehime University, for supporting protein-protein interaction analysis, and Akihiro Yano, Ehime University, for supporting chemical screening. This work was mainly supported by the Platform Project for Supporting Drug Discovery and Life Science Research (Basis for Supporting Innovative Drug Discovery and Life Science Research [BINDS]) from AMED under grant number JP21am0101077 (T.M. and T.S.), a Grant-in-Aid for Scientific Research on Innovative Areas (JP16H06579 for T.S.) from the Japan Society for the Promotion of Science (JSPS). This work was also partially supported by JSPS KAKENHI (JP19H03218 for T.S., and 19K05815 for A.N.), Takeda Science Foundation.

## Author contributions
A.N. performed the screening of chemical compounds, analysed the data, and wrote the manuscript. R.M. performed an interaction assay of GID and DELLA proteins from grapes and screening of chemical compounds. R. Hirose performed an interaction assay of GID and DELLA proteins from *Arabidopsis* and rice and analysed the effect of diphegaractin on lettuce and citrus. R. Hori and C.M. analysed the effect of diphegaractin on the interaction of GID and DELLA proteins from grapes and lettuce. T.M. and M.T. performed docking simulations of GID and diphegaractin. Y.S. analysed the effect of diphegaractin on citrus fruits. K.N. isolated the GID and DELLA cDNA. Y.H. performed the RNA sequencing and analysed the data. K.F., Y.A., and S.S. analysed the effect of diphegaractin on grapes. T.S. supervised the project, designed the study, analysed the data, acquired the funding, and wrote the manuscript. All authors contributed to the manuscript.

## Competing interests
The authors declare no competing interests.
