## [Peer Review File · Communications Biology]

Reviewers' comments:

Reviewer #1 (Remarks to the Author):

The manuscript written by Nozawa et al. reports the in vitro screening of gibberellin agonists using an exquisite system that can monitor the interaction of two proteins. The screening system itself has been reported, and the activity of the agonist diphegaractin seems to be weak, not sufficient for use. So, unfortunately, I don't think it's worth appearing on Comm. Biol.

I think the manuscript involves two significant problems.

(i) In Extended Data Figure 4, the authors showed the clear interaction of the Arabidopsis two proteins with the application of 100 μ M gibberellin A3, whereas almost no response with 100 μ M diphegaractin. Nevertheless, in the gene expression analysis shown in Figure 5, both gibberellin A3 and diphegaractin were used at 50 μ M (see Methods, line 436). Under the condition, we can hardly expect the gibberellin-related signal transduction in Arabidopsis to occur with the application of 50 μ M diphegaractin.

(ii) Diphegaractin induced the interaction of the two rice proteins (Extended Data Figure 3), whereas the interactions derived from Arabidopsis are not very clear (Extended Data Figure 4). From these results, the authors described that diphegaractin may or may not function as a gibberellin agonist (see lines 175-176). If so in Fig. 6, the evaluation of diphegaractin's effect should be limited to grapes, because any interaction of the two proteins derived from other plants besides grapes is not shown.

Reviewer #2 (Remarks to the Author):

In this paper, the authors report a novel agonist for grape GA receptor using a GID1-DELLA interaction detection system based on a wheat cell-free and AlphaScreen technologies. This screening system could be accomplished within two days, including the preparation of the proteins, suggesting that it is very useful. Furthermore, the agonist found in this study showed GA-like activity not only in grapes, but also in lettuce, Arabidopsis and citrus. In this context, the identification of new GA agonists will provide new insights as plant growth regulators for agricultural crops, I believe. Unfortunately, however, the paper contains several concerns and problems. I have pointed these out as follows.

1) Figures 3 and 4 show that the size of the alkyl group of diphegaractin contributes to the high binding affinity to I/F type residues of VvGID1b. What kind of binding is this interaction between VvGID1b and the diphegaractin? As the size of the alkyl group becomes even longer, does these interaction activities declined by steric constraint?

2) In Figure 5, RNA sequencing analyzed to confirm the effect of diphegalactin on gene expression. What about the expression of GA-related genes (e.g. GA metabolic and GA biosynthetic genes)? GA concentrations are regulated by the expression levels of GA metabolic and GA biosynthetic genes. It would be important to verify whether these genes respond to agonist as well as GA4.

3) Extended Data Fig. 7a shows that the root growth promotes by diphegaractin in Arabidopsis. However, it has been reported that the rescue effect of GA4 on root elongation is considerably stronger than that on aboveground organs (Yoshida, H. et al. Proc Natl Acad Sci U S A. (2018) 115: E7844-E7853). It shows that the gid1b mutant was affected more significantly than the others. Tanimoto discussed that preferential root growth in eudicot plants, including *A. thaliana* and *L. sativa*, may depend on a difference in GA sensitivity between roots and shoots under low GA condition (Tanimoto, E. et al. Ann. Bot. (2012) 110:373-381). Therefore, it is necessary to discuss such studies.

Miner point

1) Figure 6f does not have a scale bar.

Point-by-Point Responses to the Reviewers' Critiques

We deeply appreciate the thorough analysis and constructive suggestions provided by the two reviewers to further improve our manuscript. As described in more detail below, we have experimentally addressed all the reviewers' concerns. With this extensive revision, we hope that the reviewers will concur with us that we have addressed all of the raised concerns in a satisfactory manner and, consequently, substantially strengthened our paper.

Reviewer #1 (Remarks to the Author):

The manuscript written by Nozawa et al. reports the in vitro screening of gibberellin agonists using an exquisite system that can monitor the interaction of two proteins. The screening system itself has been reported, and the activity of the agonist diphegaractin seems to be weak, not sufficient for use. So, unfortunately, I don't think it's worth appearing on Comm. Biol.

We wish to express our appreciation to the reviewer for his/her insightful comments, which have helped us significantly improve the paper.

I think the manuscript involves two significant problems.

(i) In Extended Data Figure 4, the authors showed the clear interaction of the Arabidopsis two proteins with the application of 100 μ M gibberellin A3, whereas almost no response with 100 μ M diphegaractin. Nevertheless, in the gene expression analysis shown in Figure 5, both gibberellin A3 and diphegaractin were used at 50 μ M (see Methods, line 436). Under the condition, we can hardly expect the gibberellin-related signal transduction in Arabidopsis to occur with the application of 50 μ M diphegaractin.

Answer 1.

We thank you for your critical comments for the improvement of our manuscript. We examined experimental condition for AlphaScreen and reanalyzed effect of GA₃ and diphegaractin for interaction of GID1 proteins with DELLA proteins from Arabidopsis. In this experiment, we could get a little bit higher activity of diphegaractin. So, we replaced Extended Data Fig. 4b

with the new data. Furthermore, we tested concentration dependent activity of diphegaractin for facilitation of interaction of AtGID1b with RGA. The result was shown in Extended Data Fig. 4c. In this experiment, approximately 10^5 of AlphaScreen signal value was detected at 50 μM of diphegaractin. We think this AlphaScreen signal value of diphegaractin is lower than that of GA₃ but that value is sufficient for eliciting signal transduction of gibberellin. In Fig. 6d, we showed promotion activity of diphegaractin for root elongation in lettuce at 50 μM . In this revised manuscript, we tested promotion activity of diphegaractin at 50 μM for interaction between GID1 and DELLA from lettuce, and detected 5×10^4 of AlphaScreen signal value at combination of LsGID1b-1/LsDELLA1 and LsGID1b-2/LsDELLA1 (Extended Data Fig. 5b).

(ii) Diphegaractin induced the interaction of the two rice proteins (Extended Data Figure 3), whereas the interactions derived from Arabidopsis are not very clear (Extended Data Figure 4). From these results, the authors described that diphegaractin may or may not function as a gibberellin agonist (see lines 175-176). If so in Fig. 6, the evaluation of diphegaractin's effect should be limited to grapes, because any interaction of the two proteins derived from other plants besides grapes is not shown.

Answer 2.

We thank you for your critical comments for the improvement of our manuscript.

Results from rice and Arabidopsis (Extended Data Fig. 3 and 4) showed that diphegaractin promoted the interaction of some combination of GID1 and DELLA proteins from Arabidopsis but the promotive activity was negligibly in rice. So, we thought diphegaractin might have GA agonist activity in some plants but not in other plants. In this revised manuscript, we tested GID1-DELLA interaction-promoting activity using GID1 and DELLA proteins from lettuce and detected promoting activity of diphegaractin in some combination of GID1 and DELLA proteins from lettuce. As root elongation activity of diphegaractin was already shown in Fig. 6d, these results indicate that diphegaractin should have GA agonist activity in some plants at some organs. Hence, we added the data of effect of diphegaractin for in vitro interaction analysis of GID1 and DELLA proteins from lettuce as Extended Data Fig. 5 in the revised manuscript and replaced sentences in line 171-178 as following sentences.

Next, to determine whether diphegaractin shows GA agonist activity in other plant GA receptors, we analyzed interaction-promoting activity using GID1 and DELLA proteins from

rice, Arabidopsis, and lettuce. Interestingly, although diphegaractin activated negligibly the interaction of OsGID1-OsSLR (Extended Data Fig. 3), it promoted the interaction of AtGID1b with AtRGA, AtGID1c with AtRGA and AtGAI, LsGID1b-1 with LsDELLA1, and LsGID1b-2 with LsDELLA1 (Extended Data Fig. 4 and 5). These results indicate that diphegaractin is expected to have GA agonist activity at particular set of GID1 and DELLA proteins in some plants.

Reviewer #2 (Remarks to the Author):

In this paper, the authors report a novel agonist for grape GA receptor using a GID1-DELLA interaction detection system based on a wheat cell-free and AlphaScreen technologies. This screening system could be accomplished within two days, including the preparation of the proteins, suggesting that it is very useful. Furthermore, the agonist found in this study showed GA-like activity not only in grapes, but also in lettuce, Arabidopsis and citrus. In this context, the identification of new GA agonists will provide new insights as plant growth regulators for agricultural crops, I believe. Unfortunately, however, the paper contains several concerns and problems. I have pointed these out as follows.

We wish to express our appreciation to the reviewer for his/her kind comments, which have helped us significantly improve the paper.

1) Figures 3 and 4 show that the size of the alkyl group of diphegaractin contributes to the high binding affinity to I/F type residues of VvGID1b. What kind of binding is this interaction between VvGID1b and the diphegaractin ? As the size of the alkyl group becomes even longer, does these interaction activities declined by steric constraint?

Answer 1.

We thank you for your thoughtful comments for the improvement of our manuscript. According to your comment, we analyzed the interactions between VvGID1b and diphegaractin using docking model. In addition, we have conducted the docking simulation of the diphegaractin analogous compounds with a longer alkyl group than diphegaractin. Hence, we added figures

presenting the interactions between VvGID1b and diphegaractin and docking models of diphegaractin analogous compounds as Extended Data Fig. 8 and 9 and following sentences explaining those models in line 225-245.

Furthermore, we have conducted the docking simulation of the diphegaractin analogous compounds with a longer alkyl group than diphegaractin, 2,2-diphenyloctanoic acid (D-C-8), 2,2-diphenynonanoic acid (D-C-9), and 2,2-diphenyldecanoic acid (D-C-10). All the tested compounds showed the similar binding pose to diphegaractin toward the GA-binding pockets of OsGID1 (Extended Data Fig. 8). VvGID1a was also presumed to have a GA-binding pocket suitable to bind diphegaractin homologous compounds with a long acyl group since DC8 and DC10 showed binding poses similar to diphegaractin for VvGID1a. In contrast, there is no binding pose of the tested compounds observed in the GA-binding pocket of VvGID1b. We analyzed the size of the GA-binding pockets using CASTp 3.0 server²¹, suggesting that the pocket size of VvGID1b (169 Å³) was smaller than those of OsGID1 (208 Å³) and VvGID1a (173 Å³). Depending on the shape of the GA-binding pocket, homologous compounds with longer alkyl groups than diphegaractin are thought to cause steric hindrance to the GA-binding pocket of VvGID1b. Finally, we analyzed detail arrangement of diphegaractin in VvGID1b using docking model (Extended Data Fig. 9). Based on this binding model, I/F type residues form hydrophobic interactions with diphenyl group and a part of alkyl group of diphegalactin. The alkyl group is arranged to fill the space enclosed by I/F type residues and other residues located at the bottom of the GA-binding pocket. The binding model also suggests that S115 and R243 contribute to hydrogen bond formation with the carboxy group of diphegalactin.

2) In Figure 5, RNA sequencing analyzed to confirm the effect of diphegalactin on gene expression. What about the expression of GA-related genes (e.g. GA metabolic and GA biosynthetic genes)? GA concentrations are regulated by the expression levels of GA metabolic and GA biosynthetic genes. It would be important to verify whether these genes respond to agonist as well as GA4.

Answer 2.

We sincerely thank you for raising this concern. In response to your comment, we analyzed the data of RNA sequencing and added following sentences explaining the effect of diphegaractin for expression of GA metabolic and GA biosynthetic genes in line 279-290.

We next investigated effect of diphegaractin for expression of GA synthetic and GA metabolic genes. In this RNA sequencing experiment, GA synthetic genes downregulated (\log_2 fold change < -1) by GA₃-treatment were AtGA20ox2 (-3.9), AtGA20ox3 (-2.3), AtGA3ox1 (-2.0), and AtGA3ox2 (-1.6) and GA metabolic genes upregulated (\log_2 fold change > 1) by GA₃-treatment were AtGA2ox2 (1.1) and AtGA2ox6 (2.9). In these genes, AtGA20ox3 was also similar level downregulated by diphegaractin (-2.3), indicating feedback regulation of AtGA20ox3 by GA would be mediated by AtGID1b/AtRGA, AtGID1c/AtRGA, and/or AtGID1c/AtGAI. Whereas, expression level of AtGA20ox2, AtGA3ox1, AtGA3ox2, and AtGA2ox6 were hardly affected by diphegaractin-treatment ($1 > \log_2$ fold change > -1). In contrast, AtGA2ox2 was upregulated by GA₃ (1.1) but downregulated by diphegaractin (-3.0). The reason of this difference needs further investigation.

3) Extended Data Fig. 7a shows that the root growth promotes by diphegaractin in Arabidopsis. However, it has been reported that the rescue effect of GA₄ on root elongation is considerably stronger than that on aboveground organs (Yoshida, H. et al. Proc Natl Acad Sci U S A. (2018) 115: E7844-E7853). It shows that the *gid1b* mutant was affected more significantly than the others. Tanimoto discussed that preferential root growth in eudicot plants, including *A. thaliana* and *L. sativa*, may depend on a difference in GA sensitivity between roots and shoots under low GA condition (Tanimoto, E. et al. Ann. Bot. (2012) 110:373–381). Therefore, it is necessary to discuss such studies.

Answer 3.

We thank you for bringing up these constructive comments. According to your comment, we cited manuscripts of Yoshida et al. and Tanimoto and added following sentences to the discussion in line 350-360.

In Arabidopsis, AtGID1b was reported to express higher level in roots (Griffiths et al., 2006). In addition, root growth of AtGID1b mutant was inhibited by lower concentration of Ancyimidol, an inhibitor of GA synthesis, than that of AtGID1a and AtGID1c²³. Furthermore, lettuce was reported to have high GA sensitivity in roots²⁴, and LsGID1b-1 and LsGID1b-2 were also reported to express preferentially in

roots^{23,25}. These results indicated AtGID1b, LsGID1b-1, and LsGID1b-2 contribute GA-induced root elongation. In this study, diphegaractin promoted interaction of AtGID1b, LsGID1b-1, and LsGID1b-2 with DELLA protein and root growth in Arabidopsis and lettuce (Extended Data Fig. 4b, 10a). These results suggest that AtGID1b, LsGID1b-1, and LsGID1b-2 would be mainly contributed to GA-induced root growth in Arabidopsis and lettuce, respectively.

Minor point

1) Figure 6f does not have a scale bar.

Answer 4.

We thank you for bringing up these constructive comments. In response to your comment, we added a scale bar at Fig 6a and f.

Reviewers' comments:

Reviewer #1 (Remarks to the Author):

I made two points about this manuscript's first version (R0). In response to my comments, the authors performed additional experiments, which yielded good results. However, the following points still need to be sufficiently explained, so I would appreciate it if the authors could provide transparent information.

(1) Using Arabidopsis DELLA and GID1, the authors reanalyzed the effects of GA3 or diphegaractin in AlphaScreen. Compared to the Extended Data Fig. 4b at version R0, the result of the Extended Data Fig. 4b at version R1 shows that all AS signals with the addition of GA3 are generally smaller. In contrast, all AS signals with the addition of diphegaractin are larger. If both effects of GA3 and diphegaractin increased, it could be explained by the increased detection sensitivity of AlphaScreen. However, I would like the authors to explain what method improvements were made regarding the difference in the increase and decrease in the AS signal values between GA3 and diphegaractin.

(2) In the comments of the authors who reanalyzed AlphaScreen using DELLA and GID1 of *A. thaliana*, "About 10(5) AlphaScreen signal was detected by adding 50 μ M diphegaractin. The value is sufficient to induce GA signal transduction." I think it would be better to show any evidence for this. In other words, what does the AlphaScreen signal value reflect, and what value is physiologically meaningful (*)?

In addition to this, version R1 adds the results of AlphaScreen using lettuce DELLA and GID1 (Extended Data Fig. 5b). As shown in Figure 6d; the authors observed approximately the same activity of lettuce root elongation between 0.1 μ M GA3 and 50 μ M diphegaractin. On the other hand, the authors showed the results by AlphaScreen based on the conditions where 50 μ M of both GA3 and diphegaractin. So, there is no standard for detecting the appropriate level of AS signal. With the addition of diphegaractin, very faint AS signals were detected in the combination of LsGID1b-1&LsDELLA1 and LsGID1b-2&LsDELLA1. I think providing some information regarding the above (*) points is necessary.

Reviewer #2 (Remarks to the Author):

The authors have responded properly to my comments and suggestions.

However, in Answer 2, the expression levels of the most GA synthetic and GA metabolic genes were not affected by diphegaractic-treatment. Therefore, it does not seem that diphegaractic has the similar biological functions as GA3.

Minor point

Line 223, is "VvGID1b (173 Å3)" a mistake for "VvGID1a"?

Point-by-Point Responses to the Reviewers' Critiques

We deeply appreciate the thorough analysis and constructive suggestions provided by the two reviewers to further improve our manuscript. As described in more detail below, we have addressed all the reviewers' concerns. With this revision, we hope that the reviewers will concur with us that we have addressed all of the concerns in a satisfactory manner and, consequently, substantially strengthened our paper.

Reviewers' comments:

Reviewer #1 (Remarks to the Author):

I made two points about this manuscript's first version (R0). In response to my comments, the authors performed additional experiments, which yielded good results. However, the following points still need to be sufficiently explained, so I would appreciate it if the authors could provide transparent information.

We wish to express our appreciation to the reviewer for his/her insightful comments, which have helped us significantly improve the paper.

(1) Using Arabidopsis DELLA and GID1, the authors reanalyzed the effects of GA3 or diphegaractin in AlphaScreen. Compared to the Extended Data Fig. 4b at version R0, the result of the Extended Data Fig. 4b at version R1 shows that all AS signals with the addition of GA3 are generally smaller. In contrast, all AS signals with the addition of diphegaractin are larger. If both effects of GA3 and diphegaractin increased, it could be explained by the increased detection sensitivity of AlphaScreen. However, I would like the authors to explain what method improvements were made regarding the difference in the increase and decrease in the As signal values between GA3 and diphegaractin.

Answer 1.

We thank you for your thoughtful comments. In this case, we obtained higher AlphaScreen signals in diphegaractin samples by using smaller volume of proteins. In first version, 0.5 μ l Arabidopsis GID1 and DELLA proteins were used for AlphaScreen Assay. Whereas 1 μ l proteins were used in revised version.

(2) In the comments of the authors who reanalyzed AlphaScreen using DELLA and GID1 of *A. thaliana*, "About 10(5) AlphaScreen signal was detected by adding 50 μ M

diphegaractin. The value is sufficient to induce GA signal transduction." I think it would be better to show any evidence for this. In other words, what does the AlphaScreen signal value reflect, and what value is physiologically meaningful (*)?

Answer 2.

We thank you for your critical comments. In order to answer to your comment, we add following sentences in line 173-177.

In our previous study, when a chemical, lenalidomide, which induces interaction of CRBN and its target PLZF, was used at concentrations where 20-fold or more signals than negative control was detected in AlphaScreen, we detected interaction of CRBN and PLZF in vivo (cited as no.17 in revised manuscript, Yamanaka, S., et al., EMBO J. 2021). Hence, we considered that the signals 20-fold or more than negative control is physiological meaningful in AlphaScreen. Using this criterion,

In addition to this, version R1 adds the results of AlphaScreen using lettuce DELLA and GID1 (Extended Data Fig. 5b). As shown in Figure 6d; the authors observed approximately the same activity of lettuce root elongation between 0.1 μM GA3 and 50 μM diphegaractin. On the other hand, the authors showed the results by AlphaScreen based on the conditions where 50 μM of both GA3 and diphegaractin. So, there is no standard for detecting the appropriate level of AS signal. With the addition of diphegaractin, very faint AS signals were detected in the combination of LsGID1b-1&LsDELLA1 and LsGID1b-2&LsDELLA1. I think providing some information regarding the above (*) points is necessary.

Answer 3.

We thank you for your critical comments. As this comment is same type of question of comment (2), please see our answer to the comment (2).

Reviewer #2 (Remarks to the Author):

The authors have responded properly to my comments and suggestions.

We wish to express our appreciation to the reviewer for his/her kind comments, which have helped us significantly improve the paper.

However, in Answer 2, the expression levels of the most GA synthetic and GA metabolic genes were not affected by diphegaractic-treatment. Therefore, it does not seem that diphegaractic has the similar biological functions as GA3.

Answer 1.

We thank you for your thoughtful comments. We agree with your comment. We replaced the sentence “The reason of this difference needs further investigation.” with the sentences “These results indicate that the expression levels of most of GA synthetic and metabolic genes seem not to be affected by diphegaractin. The response of these genes to GA would be mediated by the combinations of GID1 and DELLA proteins except for AtGID1b/AtRGA, AtGID1c/AtRGA, and AtGID1c/AtGAI.” in line 278-281.

Minor point

Line 223, is “VvGID1b (173 Å³)” a mistake for “VvGID1a”?

Answer 2.

We sincerely thank you for pointing out our mistake. According to your comment, we replaced VvGID1b with VvGID1a.

REVIEWERS' COMMENTS:

Reviewer #1 (Remarks to the Author):

I think the authors have responded appropriately to my comments.